# *Halophila Balfourii* Solereder (Hydrocharitaceae)—An Overlooked Seagrass Species

**DOI:** 10.3390/plants9111614

**Published:** 2020-11-20

**Authors:** John Kuo

**Affiliations:** Centre for Microscopy, Characterisation and Analysis, The University of Western Australia, 35 Stirling Highway, Perth, WA 6009, Australia; john.kuo@uwa.edu.au

**Keywords:** *Halophila balfourii* Solereder, *Halophila stipulacea* (Forrsk.) Asch., papillose leaf epidermal cells, seed coat surface, microscopy

## Abstract

*Halophila balfourii* Solereder has long been treated as a synonym of *Halophila stipulacea* (Forrsk.) Asch., although it was named more than a century ago. Microscopic (optical microscope and scanning electron microscope) studies on all available herbarium materials of these two species have reconfirmed that the unique papillose leaf epidermis is only presented in *H. balfourii* but not in *H. stipulacea*. The pattern of seed testa reticulate is significantly different between these two species. Furthermore, *H. balfourii* is predominately restricted to the Rodriguez and Mauritius Islands while membranous leafed *H. stipulacea* is widely distributed in the Red Sea, the Indian Ocean and the Mediterranean Sea as well as East Africa coasts. Based on distinctive characteristics of the leaf and seed coat, and its geographic distribution, it is recommended to reinstate *H. balfourii* as an independent species and not as a synonym of *H. stipulacea*.

## 1. Introduction

*Halophila* plants are small in size and their reproductive structures, such as flowers and fruits, are simple and rarely collected. As a result, the identification characteristics of *Halophila* species are rather limited and are mainly based on gross vegetative morphology [1,2]. Consequently, the nomenclatural confusion has been continuously present and the names used in the literature may often be changed or misapplied. Although Tomlinson [3] stressed that sufficient microscopic detailed features of *Halophila*’s vegetative anatomy could be applied allowing a clear separation of taxa, such valuable characteristics have hardly been used.

Currently, *Halophila stipulacea* (Forrsk.) Asch. contains two synonyms: *H. bullata* (Délile) Asch. and *H. balfourii* Solereder [1]. *H. stipulacea* was originally named by Forrskål [4] as *Zostera stipulacea* from Mocha, Yemen, with a brief description. A similar seagrass from the Red Sea was named and illustrated by Délile [5] as *Zostera bullata*. These two species were transferred to either the genus *Thalassia* or *Barkania* by various botanists before Ascherson [6] finally settled on them as a single species—*Halophila stipulacea* (Forrsk.) Asch.

Balfour [7] found two *Halophila* species, *H. ovalis* and *H. stipulacea*, growing on the reefs of the Rodriguez Islands while he was participating in “The Transit of Venus Expeditions, 1874–75”. Balfour [8] reported in great detail the vegetative morphology, anatomy and reproductive biology of these two *Halophila* species. Through a microscopic study on these *Halophila* species, Balfour commented and illustrated specifically that *H. stipulacea* from the Rodriguez Is. had a remarkable leaf surface with a papillose or tuberulated appearance ([8] Plate lX, Figure 21). Ascherson and Gürke [9] and Ascherson [10] did not discuss this unusual leaf epidermal morphology when they redescribed *H. stipulacea*. On the other hand, based on this unique morphological characteristic (the papillose or tuberulated appearance), Solereder [11] introduced a new species name, *Halophila balfourii*, for the *H. stipulacea* from the Rodriguez Is. without a formal species description. However, Ostenfeld [12], Ostenfeld and Meerestraser [13], den Hartog [1] and Simpson [14] continued treating *H. balfourii* as well as *H. bullata* as synonyms for *H. stipulacea* and this classification has been maintained in the recent seagrass taxonomic reviews [2,15]. Leaves of *H. stipulacea* have been collectively described as cartilaginous to membranous; surface glabrous, papillose or with minute hairs, occasionally bullate [1,2]. In the meanwhile, *H. stipulacea* had exhibited the migrating capability to expand into the Mediterranean Ocean from the Red Sea after the opening of the Suez Canal [16,17] and then, recently, to the Eastern Caribbean [18,19,20,21,22].

Waycott et al. [23] initiated a molecular genetic survey on the *Halophila* taxonomy by using the *H. stipulacea* materials from the Mediterranean Ocean and concluded that *H. stipulacea* is a distinct *Halophila* species. Nguyen et al. [24] had also molecularly identified *H. stipulacea* from the Chilika Lake, India (19°43′ N; 85°19′ E). Unfortunately, no molecular genetic or microscopic study has been undertaken on the species status of *Halophila balfourii* since the species was named more than a century ago.

The present microscopic examination on all available herbarium materials of *H. stipulacea*, *H. balfourii* and *H. bullata* from a wide geographic area aims to reconfirm the papillose epidermal appearance of *H. balfourii* in order to reinstate the taxonomic status of *H. balfourii* as an independent species instead of as a synonym to the *H. stipulacea*.

## 2. Results and Discussion

### 2.1. Comparative Microscopic Studies on the Leaf and Seed Surfaces in H. stipulacea and H. balfourii

The results of microscopic studies on herbarium materials confirm that the *H. stipulacea* plants with bullated leaves, and occasionally with hairs, are restricted to the Red Sea; while the *H. stipulacea* plants with membranous leaves without hairs (Figure 1A) are widely distributed in the Red Sea, Mediterranean Ocean, East Africa and the Indian Ocean. On the other hand, plants with the papillose leaves of *H. balfourii* have come from the Rodriguez and Mauritius Islands (eleven collections) with only one collection coming from southern India (Palk Bay, 9°35′ N; 79°15′ E). It should be noted that the other three collections from the Palk Bay examined in this study were identified to be *H. stipulacea*.

The individual squamous epidermal cells in these *Halophila stipulacea* leaves can easily be identified (Figure 1B). The epidermal cells of the papillose leaves have an enlarged protrusion which obstructs the view to the base of epidermal cells (Figure 1C), thus each epidermal cell outline cannot be distinguished easily from the surface of the leaf (Figure 1D). Furthermore, cross sections of the membranous leaves of *H. stipulacea* possess a smooth leaf surface with a squamous appearance of epidermal cells (Figure 1E). Conversely, those of the papillose leaves of *H. balfourii* exhibit an undulated surface due to a pyramid appearance of epidermal cells (Figure 1F). This special feature of the leaf epidermal cells is unique to this species, when compared to all known seagrass species (Kuo, per. obs.). In addition, the pattern of seed testa reticulate is quite different in these two *Halophila* species; in *H. stipulacea* it appears to be square (Figure 1G) and in *H. balfourii* it shows a rectangular shape (Figure 1H).

The present microscopic study confirms Balfour’s earlier observations that the unique papillose leaves only belong to *H. balfourii* and not to *H. stipulacea*. This study further demonstrates that seed testa appearances are also quite different between these two species. In addition, *H. balfourii* is predominately restricted to the Rodriguez and Mauritius Islands. Thus, it is recommended that *H. balfourii* should have an independent species status and not be a synonym of *H. stipulacea*. Since there is no formal species description of *H. balfourii* existing, a proper taxonomic description is provided below.

It is anticipated that the present microscopic results will stimulate and encourage more molecular genetics, ecological and biological studies to be carried out on this little-known *H. balfourii* species in the near future.

### 2.2. Redescription of Halophila Balfourii

***Halophila balfourii*** Solereder *Beih. Bot. Central* l. 30, 1 (**1913**): 47. [Figure 2]*Halophila stipulacea* (Forrsk.) Asch. Balfour (1879b): 290; Ascherson (1906): 393 *pro parte*; Ostenfeld (1918): 11, *pro parte*; Ostenfeld & Meerestraser (1927): 37 *pro parte*; den Hartog (1970): 258, *pro parte*; Simpson (1989): 28, *pro parte*; Kuo & den Hartog (2001): 51, *pro parte.***TYPE:** Rodriguez Islands, Aug.-Dec 1874, *Dr IB Balfour*, holo: K; iso: C, L, NHM, P, *fide* Ferrer-Gallego & Boisset, *Taxon*, **2015**, *64*, 1035.

Marine, submerged herbs, rhizomes (0.2-) 0.5–0.8 (1.2) mm diameter, internodes (5-) 10–30 mm long, with one root at least 60 mm long at each node. Scales two, elliptic or obovate, white or transparent, glabrous, margins entire; the petiole scale, apex retuse, base ovate to oblong, subtruncate, 8–10 mm long, 2.5–3 mm wide, the rhizome scale, apex retuse, base oblong, truncate, convolute, hyaline 10–12 mm long, 2–3 mm wide. Petioles, (3-) 5–10 mm long. Leaf blade distinct elliptic, cartilaginous, papillose, not membranous, nor bullate (15-) 20–30 (-35) mm long and (1.5-) 2–4 (-5) mm wide, L:W ratio 7–12:1; apex distinctly pointed, base cuneate, symmetrically into the petioles. Margin serrulate, especially on the apical region; cross veins (6-) 10–12 (-14), unbranched; ascending at angles about 45°. Space between lamina margin and intramarginal veins, extremely narrow, up to 0.1 mm. Spathe ovate, acute, keeled. Male flowers: scales 4 mm long, 2.5 mm wide; pedicel thin, up to 10 mm long, 1 mm diameter at anthesis; tepals 3–4.5 mm long, 2–2.5 mm wide. Female flowers: ovary ovoid to ellipsoid, 1.5–2 mm long, 1–1.5 mm wide; hypanthium 4–5 mm long; styles 3, 15–25 mm long. Fruits ellipsoid (2-) 3.5–4 mm long, (2-) 3 mm wide. Seeds (12-) 20–30, globular, 0.75 × 0.5 mm, at both ends contracted, testa reticulate.


**Material examined**


**Mauritius:** Grand River, NW, Oct. 1869, *fl., N Pike 13* (C, K). Fort George, Port Lorius Harbor, 1871, *Col N Pike s.n.* (NY). Grand Bay, Oct 1929, *mf, fl, fr*, *Th Mortensen s.n.* (C, K, NHM); *loc. cit.,* Dec 1973, seeds, *Coode 4298* (K, P). Cassis, Jan 1947, *G Morin s.n.* (NHM). Isle de Ambrei, May 1868, *Mrs Morris s.n.* (NHM). Canoniers Pointe, Oct 1929, *Th Mortensen s.n.* (NHM). **Rodriguez:** No exact locality, Aug.-Dec 1874, *Dr IB Balfour s.n.* (C, K, L, NHM, P). Lagoon Face, July 1970, *Coode 2747* (P). Gravier, Dec 1973, *Coode 4341* (K). Anse Aux Anglais, Mar 1978, *Bosser 22,400* (P). **India:** Pamban, Oct 1922, *M.O.P. Iyengar 133A* (K, NHM).

**Distribution:** Currently, *Halophila balfourii* is confined to the Rodriguez and Mauritius Islands. It is very possible that *Iyengar 133 A* (K, NHM) from Pamban, India was not a true locality. Ramamurthy et al. [25] did not mention the papillose leaves of *H. stipulacea* from the Coromandel Coast. *H. balfourii* also is not present at Kenya coast, as reported by Issac previously [26].

**Habitats:** The species had been collected from less than 1 to 6 m in water depth.

**Biology:** The limited herbarium specimens indicate that both male and female flowers occur in October and fruits are produced from October to December in Mauritius.

## 3. Material and Methods

Herbarium samples of *Halophila stipulacea* (157 collections) and *H. balfourii* (12 collections) at major herbaria e.g., BSI (4), C (9), K (21), L (17), NHB (27), NY (5), P (18), S (7), UC (2), US (6), W (6), and at Tel Aviv University (80) and the Hebrew University of Jerusalem (10) were studied. The geographic distribution of these herbarium specimens is presented in the Table 1. Leaves and seeds were chosen and imaged by microscope.

Up to 60% of these herbarium specimens were imaged and measured for root lengths, rhizome internodes, petiole length, leaf length and width as well as the number of cross veins. When available, reproductive structure was measured and noted. Botanical drawings of reprehensive specimens were also made.

Only five out of 157 herbarium specimens had exhibited ‘bullata’ appearance (=*H. bullata*), apart from this feature, there are no other morphological differences from *H. stipulacea*. Therefore, these bullata specimens have been treated as *H. stipulacea* in this study.

Most of the membranous and bullated leaved *H. stipulacea* are easily identified under the dissecting microscope, however, those leaves which were covered by microscopic epiphytes were examined by SEM to confirm that these specimens were not *H. balfourii*. On the other hand, all twelve *H. balfourii* collections have been further confirmed by the SEM examination of their special papillose characters.

More than twenty selected *H. stipulacea* (with bullate, membranous and hairy appearances) representing all their geographic distributions, along with all twelve *H. balfourii* collections, were prepared for SEM imaging. It should be noted that both surfaces of each leaf were studied using a Philips Scanning Electron Microscope 505 at 15 kV.

Three samples each from *H. stipulacea* and *H. balfourii* were prepared for histology studies. Transverse sections (2 µm in thickness) on glycol methacrylate embedded samples of selected leaves were stained with 0.5% toluidine blue O (pH 4.4) and photographed using a Zeiss research light microscope.

Three seeds of *H. stipulacea* (all from Elate Gulf, Tel Aviv herbarium) and two seeds of *H. balfourii* from Rodriguez (K) were examined using the Zeiss Scanning Electron Microscope 505 at 15 kV.

## Figures and Tables

**Figure 1 plants-09-01614-f001:**
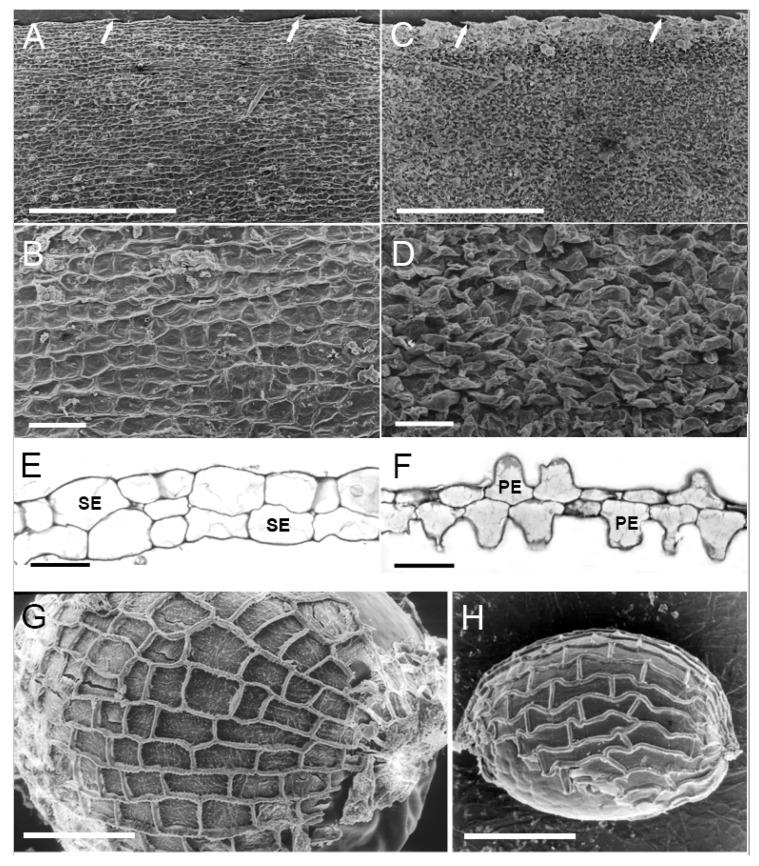
(**A**,**B**); Scanning Electron micrographs of the membranous blade surfaces of *Halophila stipulacea.* Scale bars: Figure (**A**) = 1 mm; Figure (**B**) = 1 µm. Arrows shows serrulated blade margins. Kenya, Shimo la Jewa, in pools, 10 Oct. 1965, *FM Issac A108* (NHM). (**C**,**D**); Scanning Electron micrographs of the papillose leaf blade surfaces of *Halophila balfourii*. Scale bars: (**C**) = 1 mm; (**D**) = 1 µm. Arrows shows serrulated blade margins. Mauritius, Grand Bay, Oct 1929, *Th Mortensen s.n.* (NHM). (**E**,**F**). Transverse sections of the membranous leaf blade of *Halophila stipulacea* (**E**) and the papillose leaf blade of *H. balfourii* (**F**) to show that *H. stipulacea* leaf has squamous appearance epidermal cells (SE); while *H. balfourii* leaf has pyramid appearance epidermal cells (PE). All scale bars = 1 µm. (**G**,**H**); Scanning Electron micrographs of the seed surface of *Halophila stipulacea* (**G**) and *Halophila balfourii* (**H**). Scale bars: (**G**) = 2 µm; (**H**) = 2.5 µm. *H. stipulacea*: Elate Gulf, *Lipkin 10,174* (Tel Aviv Univ); *H. balfourii*: Rodriguez, Graviers, Dec 1973, *Coode 4341* (K).

**Figure 2 plants-09-01614-f002:**
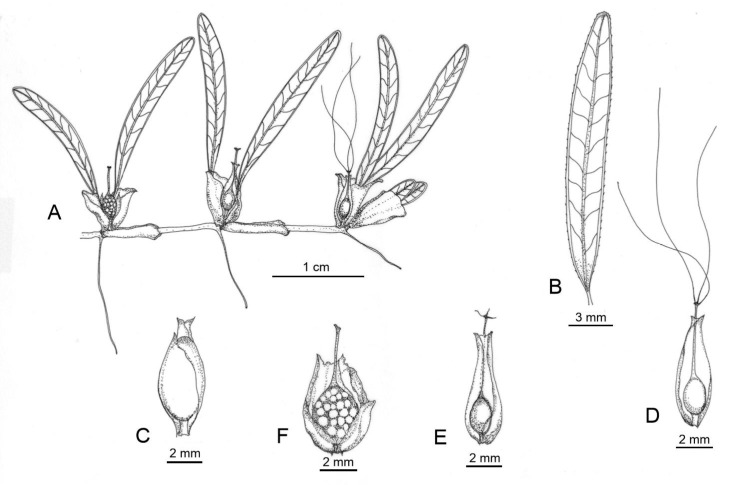
Botanical drawings of *Halophila balfourii* Solereder. (**A**) A female plant with flowers and developing fruits. (**B**) Leaf blade with unbranching cross veins. (**C**) Immature male flower. (**D**) Female flower with styles attached. (**E**) Female flower with styles detached. (**F**) Maturing fruit. Mauritius, Grand Bay, Oct 1929, ***mf, fl, fr*,**
*Th Mortensen s.n.* (C, K, NHM).

**Table 1 plants-09-01614-t001:** The geographic distribution of *H. stipulacea* and *H. balfourii* in the studied herbarium specimens (numbers).

	Red Sea	E Africa	Mediterranean Ocean	Indian Ocean	Mauritius Rodriguez
*H. stipulacea*	113	22	11	11	0
*H. balfourii*	0	0	0	1	11

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
