# Peer review of "Halophila Balfourii Solereder (Hydrocharitaceae)—An Overlooked Seagrass Species"

_plants, 2020, doi:10.3390/plants9111614_

Round 1
Reviewer 1 Report
Recommendations for author
“Halophila balfourii Solereder (Hydrocharitaceae) – A neglected seagrass species”, manuscript by John Kuo
Summary
The author John Kuo examined herbarium specimen of Halophila stipulacea, and herbarium specimen of what has previously been described as Halophila balfourii Soler. At present, H. balfourii is considered a synonym of H. stipulacea.
The author has made morphological examinations and measurements of herbarium specimen of Halophila stipulacea (138 collections) and material named H. balfourii (15 collections) under a dissecting microscope. Some selected leaf samples and seeds were analysed in scanning electron microscope and light microscope. Based on the differences found in leaf and seed surface structures, the author suggests to reinstate H. balfourii as independent species and not as a synonym or subspecies of H. stipulacea. The author gives a redescription of H. balfourii.
Comments
The author bases his recommendation to use the name H. balfourii on morphological data only. Could other information be used to support the recommendation, such as comparative DNA analysis of H. stipulacea and the material called H. balfourii? Also, it is not clear to me whether other than leaf and seed surface morphological features have been compared between the two different herbarium materials, H. stipulacea and H. balfourii. For instance, has the author compared flowers, and are there differences between flowers? A comparative description of both species could give better support for the claim to reinstall the species H. balfourii. Would it for example be possible to analyse additional fresh material in support of the recommendation to reinstall H. balfourii?
The author does not discuss whether there could be other explanations for morphological differences in leaves in the H. balfourii –material.
The author uses the name H. balfourii as if it were already considered a species on its own. To my understanding, however, the author tries to give evidence for the reinstallation of the species. Now, I find the title misleading.
I do not quite understand whether the redescription of H. balfourii is all based on specimen seen and investigated by the author.
In the Materials and Methods, it is not clear how many samples have been analysed with microscopy. Also, the number of specimen is not related to the different geographical areas.
The spelling of “Halophila balfourii” has to be checked throughout the text, because it is sometimes misspelled “balfouri”.
The spelling of Forsskål has to be checked throughout the text.
Specific comments
Title
1. lines 2-3: The title of the manuscript might have to be adjusted. As long as H. balfourii is not considered an independent species, it might be misleading to call it “a neglected seagrass species”.
Abstract
2. line 9-12: The beginning is somewhat confusing. Perhaps this sentence needs to be reframed to state first the fact and then conclusion of the research more clearly. Would it help to first state that H. balfourii is presently a synonym of H. stipulacea, and then explain what has been done in this research? The author has investigated H. stipulacea material and based on these investigations, he concludes that those material that has the unique papillose leaf epidermis is H. balfourii, and a different species than H. stipulacea. And therefore he recommends to reinstate H. balfourii as an independent species.
3. line 12: “Its distribution is restricted …” – might be better to write instead “The distribution of H. balfourii is restricted to …” Right now it is not clear to which species the author refers in this sentence.
Introduction
4. line 22: Although it is very interesting to know about the migration capability of the species H. stipulacea, the information is not really connected to the other manuscript content, because the author is not discussing migration. If H. stipulacea is effective in migrating, is it also found in Mauritius and the Rodriguez Island along with the plants called H. balfourii?
5. line 34: referring to sources should be “Ascherson and Gürke [14], and Ascherson [15]”
6. line 35: should be “when they redescribed”
7. line 37: reference Ostenfeld (1918) -> Ostenfeld [7]
8. line 37: reference Ostenfeld (1927) -> Ostenfeld and Meerestrader [17]
9. line: Some mistake with numbering the references: den Hartog is number [18], not (19) and Simpson is number [19], not (20), at least according to the reference list.
10. line 39: The author should check whether he is still referring to the right taxonomic reviews.
11. Now, it appears that the author is not referring to reference 20. in the reference list: Kuo, J.; den Hartog, C. Seagrass taxonomy and identification key. In: Global Seagrass Research Methods. Short, F.S.; Coles, R.G. (eds.) Eelsevier, Amsterdam, 2001, 31-58.
12. Lines 37-38 versus line 41: Is reference number (20) the article by Simpson, or Kuo & den Hartog?
13. line 42: What does the author mean with the sentence “The recent microscopical studies on Halophila stipulacea herbarium materials have reconfirmed Balfour’s observation of H. balfourii’s remarkable papillose structure.” The author is not specifying whose microscopical studies he is referring to? Also, the message is confusing: Is it the H. stipulacea material that shows papillose structures, and therefore, the specimen that show the papillose structures are considered to be H. balfourii?
14. lines 42-46: The author is giving here the conclusions of his study. These lines should be moved to the end of the discussion, or to a separate chapter “Conclusion”.
15. line 46: Rather: “… H. balfourii should have an independent species status and not be treated as a synonym of H. stipulacea.”
16. In the end of the introduction, the objectives of this study should be outlined: what does the study investigate and for what purpose?
Results and Discussion
17. lines 50-53: The author concludes that plants with bullate leaves are all restricted to the Red Sea, the plants with membranous leaves are widely distributed in the Mediterranean Sea, and papillose leafed plants are only known from Rodriguez and Mauritius Islands. However,
the author does not give any evidence to the reader for this statement. For example, there could be a table with morphological feature of the leaf, and numbers on how many herbarium specimen have been investigated from the respective geographical distribution area.
18. It is unclear how many herbarium specimen from the three/four different geographical areas (Red Sea, Mediterranean Sea, and Indian Ocean, and Rodriguez and Mauritius Islands, respectively) have been investigated.
19. line 62: Is the personal observation by the author (that the papillose leafed H. balfourii is only known from Rodriguez and Mauritius Islands) evidence enough to support the claim? Could this statement be connected to the total number of specimen investigated?
20. line 68: Is the year correct: Ascherson (2006)?
21. line 70: Is the year correct: Kuo & den Hartog (2005)?
22. lines 88-94: In the Material examined, eleven locations are mentioned. In line 104, it is mentioned that 15 collections of H. balfourii were studied. Apparently, five collections were not considered representative of H. balfourii? Why not?
23. lines 96-98: Could the author clarify this sentence (“Ramamurthy et al. (22) did not mention the papillose leaves of H. stipulacea …”)? What exactly is meant? If the author has himself investigated the material, was the leaf papillose or not? If on the other hand the leaf was papillose and the material from southern India, distribution might not be restricted to Mauritius and Rodriguez Islands; or might this be a sign of plasticity?
Materials and Methods
24. line 103: Material and Methods -> change to Materials and Methods
25. lines 103-114: Materials and Methods is very short.
26. line 104: Please choose a proper heading instead of using the first part of a sentence as heading. Then transfer the text in line 104 into the text body under 3.1.
27. line 104: What is meant with “collections” here? How many specimen were studied?
28. line 105: at major herbaria ...: change to -> At major herbaria …
29. line 110: Here, the author could add a sentence like e.g.: “Leaves and seeds were chosen and imaged by microscope.”
30. lines 111-112: 3.2.: The heading is missing and the text appears as heading. The author should choose a heading for what is described in this chapter and use the sentence after 3.2. as the text.
31. line 111: It is not mentioned here that also seeds were investigated in the scanning electron microscope.
32. lines 113-114: 3.3. Same comment as previous: The heading is missing and the text appears as heading. I suggest that the author should choose a heading for what is described in this chapter and use the sentence after 3.3. as the text. Please add a fullstop in the end of the sentence. I suggest to remove “samples of these selected leaves (3.2.)” and change to “samples of leaves were stained with …”.
33. lines 111-112, and 113-114: How many samples have been investigated in light and scanning electron microscope? Have all the investigated specimen shown the surface structures as shown in the figures (Figs. 1-8)?
Author Contributions
34. line 117: The author should fill in this part.
Appendices, starting line 131
35. lines 131-152: I wonder whether the figures could be integrated into the text instead of being appendices.
36. line 141, and Figures 5 and 6: What does the letter E stand for in the figures?
37. lines 147 and 149: It should be either “Botanic drawings” or “Botanical drawings”.
Typing / spelling mistakes
38. line 10: should be (Forssk.), not (Forsk.)
39. Reference to published literature at present not conform with Journal’s format. Reference numbers should be in square brackets; the author should also check how several numbers are presented (without space after comma; comma and no semicolon (compare line 39).
40. line 21: species’s – change to: species’
41. line 23: It should be “Ostenfeld”, not “Osteneld”.
42. lien 24: check spelling: it should be “Forsskål”, not Forskål. Compare also with reference list.
43. line 27: spelling of Ashcerson – it should be “Ascherson”
44. line 31: spelling of Halopphila: should be Halophila
45. line 35: uniquic ?
46. line 39: (21; 22) should be comma, not semicolon
47. line 41: (18; 20) should be comma, not semicolon
48. line 41: should be “occasionally”
49. line 58: should be “papillose”, not “papilloae”
50. line 59: should be “balfourii”, not “balfouri”
51. line 62: should be “pattern”, not “patten”
52. line 97: should be “did not mention” instead of “did not mentioned”
53. line 98: should be “balfourii”, not “balfouri”
54. line 102: should be “flowers occur”, not “flowers occurs”
55. line 106: should be “Universidade”, not “Universidale”
56. line 106: should be “Mozambique”, not “Mozanbique”
57. line 109: should be “reproductive”, not “reproduective”
58. line 110: should be “representative”, not “representive”
59. lines 135, 138, 143: Please unify whether Scanning Electron micrographs, or Scanning Electron Micrographs.
60. line 135: compare “membraneous” vs. “membranous” earlier in the text
61. line 136: should be “Arrows show”, not “Arrows shows”
62. line 138: should be “Scanning Electron micrographs”, not “Saning Electron micrographs”
63. line 138: should be “papillose leaf blade”, not “papilose leaf balde”
64. line 138: should be “Halophila balfourii”, not “Halophila balfouri”
65. line 141: compare “membraneous” vs. “membranous” earlier in the text
66. line 142: should be “papillose”, not “papilose”
67. line 142: should be “Halophila balfourii”, not “Halophila balfouri”
68. line 144: check empty space before “2.5 μm”
69. line 147: should be “Halophila”, not “Halophia”
70. line 150: check empty space between “D.” and “Female”.
71. line 174: should probably be “Vorarbeiten”
72. line 177: should be “Die natürlichen Pflanzenfamilien”
73. line 178: should be “Wilhelm Engelmann”, not “Enelmann”
74. line 180: should be “Beobachtungen”, not “Beobachtmgen”

Author Response
Response to the Reviewer 1
- General Comments:
Firstly, I really appreciate the (No. 1) reviewer’s critical, but very constructive and valuable comments and suggestions. I am also grateful that he made numerous ‘spelling’ corrections to improve the quality of my manuscript.
Unlike most terrestrial angiosperms, seagrass species identification is rarely dependent on flower structures as a criterium at species level, though these floral structures are sometimes used at higher levels’ classifications (subgenera, genera, family). For the flower structure, probably only the style numbers are different in some sections (= subgenera) in Halophila. Therefore vegetative morphological structures are mainly used for species identification in Halophila.
There are a few DNA genetics gene markers (ITS, Kmat, rbcL) that have been applied for Halophila species identification. Most of these studies only analyzed a few ‘local samples’, and rarely included the specimens from the species type locality. Therefore, these ‘molecular species identification’ results quite often become questionable. Furthermore, some gene markers are not sensitive enough or are not suitable for identifying ‘closely related species’, resulting in these species being considered as hybrids or the conspecific with H. ovalis, the most common and widely distributed Halophila species. I have included two molecular studies on H. stipulacea in the revised ms.
Solereder had named H. balfourii (1913) but did not provide the species description; therefore, based on herbarium specimens I presented a species description of H. balforurii.
I have prepared this revised ms according to Tomlinson’s recommendation to re-examine Dr. Balfour’s earlier microscopic investigation. In addition, I have used SEM to study the leaf surfaces and seed surfaces in more detail and the seed surface structure could become an additional species identification criterium in Halophila.
Following the reviewer’s valuable suggestions, I have made substantial modification to this manuscript, including to change the Title, rewriting the Introduction, expending the Materials and Methods and modifying Results and Discussion sections. I have also included a table in response to the reviewer’s request on the points 18, 22, 27.
There has not been any study on H. balfourii since it was named more than a century ago. My aim for publishing this paper is to raise the awareness of the existence of this little known but unique species to seagrass researchers in the hope that they can be encouraged to carry out more molecular genetic, ecological and biological studies on H. balfourii in the future.
- Specific Comments:
Title
- Change to a new title as “Halophila balfourii Solereder (Hydrocharitaceae) – An Overlooked Seagrass Species”. Lines 1, 2
Abstract
- I have changed the abstract by beginning with “H. balfourii …” as suggested.
Line 14
- Changed to “H. balfourii is predominately restricted ……” Line 19
Introduction
- The migration capability of H. stipulacea – have been moved to Line 60-61
- stipulacea was not found in Mauritius Is and the Rodriguez Is.
- Changed .. Line 51
- Changed as recommended. Line 52
- Changed Line 55
- Changed Line 56
- They have been corrected Line 56
- I have checked all the references.
- I have included the Kuo & den Hartog as [2] Line 33
- I have checked and corrected all references.
- I have rephrased this sentence. Line 69
- I have rephrased Lines 71,72
- I have rephrased. Lines 69-72
- I have changed as suggested. Lines 71-72
Results and Discussion
- I have included Table I to address this comment. Appendix A
- See Table I Appendix A
- Based on my personal microscopic studies, papillose leaf blades only occur in H. balfourii among more than twenty Halophila species. Similar structures do present in Heterozostera nigricalus, but not in any other seagrass species.
- It was 1906. Line 109
- It was not 2005, should be 2001 and has been corrected Line 111
- I was wrong, I only identified 12 H. balfourii collections in this study – Sorry for the mistake made.
- Ramamurthy et al. included H. stipulacea from Coromandel Coast (Palk Bay is part of Coromandel Coast). These authors did not mention ‘pappilose leaves’, indicating no H. balfourii in their study. I have examined four collections of H. stipulacea-H. balfourii from India, all from Palk Bay area, and only the Iyengar’s collection from Pamban (K), which is located in Palk Bay, was identified to be H. balfourii while the rest of three collections were found to be H. stipulacea. However, I have a deep doubt on the true collecting locality of Iyengar’s collection.
Materials and Methods
- Changed Line 146
- I have expanded this section. – Lines 146-170
- Sub-headings have been cancelled. Lines 146-170
- See Table I Appendix A
- Changed Line 148
- 29. Added this sentence as suggested Line 151
- Changed as requested Line 146-170
- Seeds SEM is mentioned. Lines 168-170
- Changed as the reviewer suggested. Lines 164-167
- More than twenty H. stipulacea and 12 H. balfourii specimens were examined with SEM and two specimens from each species were transverse sectioned for LM. The representative images were shown in Fig. 1-8.
Author Contributions
- I am the sole author of this paper and had conducted the entire research. However, in the acknowledgement, I have thanked those who have assisted in processing the specimens from the herbaria, the computer assistance and botanical drawings. Lines 172-177
Appendices
- I do agreed with this recommendation but it is subject to the Journal Editor’s decision.
- I have added E: Epidermal cells in the legend.
- Changed to Botanical drawings Lines 214-215
Typing/spelling mistakes
38-74. All have corrected. Thank You!!
Reviewer 2 Report
This paper deals with morphological differences using microscopic studies to reinstate Halophila balfourii as an independent species and not as a synonym of H. stipulacea. The genus Halophila contains approximately 20 species, based on morphological differences. Although some Halophila species often recognized as a synonym species with other Halophila species in the past, these species have currently been renamed as an independent species by various scientific methods, such as taxonomic and molecular genetic studies. This study well described for morphological differences (e.g. the surface and cross sections of the leaf and the pattern of seed testa reticulate) between H. balfourii and H. stipulacea by microscope. Therefore, it is acceptable to reinstate H. balfourii as an independent species status, as based on the methods adopted and the results obtained by the authors.
Author Response
Thank you for the positive comment and agree that Halophila balfourii Solereder should be reinstated as an independent species.
Round 2
Reviewer 1 Report
I think that the modifications that the author has made to the text have greatly clarified the manuscript. I acknowledge the author’s responses to general remarks.
Nevertheless, I still have some comments.
Recommendations for author
plants-967210-peer-review-v2, manuscript by John Kuo
Comments
I think that the modifications that the author has made to the text have greatly clarified the manuscript. I acknowledge the author’s responses to general remarks.
Nevertheless, I still have some comments.
Table 1 helps the reader understand from which areas the different H. stipulacea and H. balfourii specimen came from and confirms the geographic distribution. However, now the numbers for H. stipulacea specimens in table 1 do not match the number of “collections” as stated in line 147. I am not quite sure whether they should match, but in table 1, the sum of all H. stipulacea specimens studied is 113+22+11+11 = 157 (and not 138). I am not sure either whether “collections” and specimens are used as synonyms here. But for instance in line 155, the author writes “Only five out of 138 herbarium specimens had exhibited “bullata” appearance, …”, and in line 147, the author writes “138 collections”, which indicates that the words are used as synonyms.
What do the numbers in parentheses after the herbaria mean? (lines 147-148)
The text should still be checked for typing/spelling mistakes. The author is encouraged to check how the name of Forsskål is spelled and correct accordingly, also when the name is abbreviated. I think the name is still wrongly spelled in e.g. lines 10, 19, 30, 31, 35, 106, 179. There might be more places.
Especially the figure text (lines 93-103) has to be checked because it contains widely the same mistakes as were in the original manuscript. E.g. line 96: “Saning Electron micrographs” should be “Scanning Electron micrographs”; line 96: “papilose balde surfaces” should be “papillose blade surfaces”; line 97: “balfouri” should be “balfourii”; “Maritius” should be “Mauritius”; line 100: “papilose leaf blade of H. balfouri” should be “papillose leaf blade of H. balfourii”; line 101: “Scanninng Electron Micrographs” should be “Scanning Electron micrographs”; line 102: check space before “2.5 μm”
Even though the author states that he has added E: Epidermal cells in the legend, this is not the case. The letter E is still not explained in the figure legend (compare line 99).
After adjusting the Materials and Methods section, unfortunately, the author has deleted “Transverse sections (2 μm in thickness) on glycol methacrylate embedded samples of these selected leaves were stained with toludine blue O and photographed using a Zeiss research light microsocpy”. (Compare lines 113-114 in the first version of the manuscript.) Was this deletion intentional? Should preparation of the samples anyhow be mentioned? Perhaps in line 166?
Also, now there is no more mentioning of the author’s measurements on the herbarium specimen (compare with lines 107-110 in the first version of the manuscript). Was this deletion intentional?
Additional typing/spelling mistakes:
line 23: to my understanding “productive structures” should be “reproductive structures”
line 44: “balforurii” should be “balfourii”
line 60: “a wide geographic areas” should be “a wide geographic area”
line 142: “Halophila balfouri” should be “Halophila balfourii” with two times letter “i”
line 144: “Maritius” should be “Mauritius”?
line 146: should be “Materials and Methods”; at least according to the instructions for authors
line 185: “Die natürlichen. Pflanzenfamilin” should be “Die natürlichen Pflanzenfamilien”. And no full stop after “natürlichen”
Author Response
Many thanks again to Reviewer 1’s further comments and queries on the previous revised manuscript, particularly in relation to the ‘E’ in the figure legends (Figs. 5 and 6). My responses to these comments and queries are as follows:
- The total herbarium specimens of H. stipulacea (including H. bullata) should be 157 as shown in Table 1. This number has replaced the wrong number 138 at Lines 147, 157.
- I have deleted those numbers in parentheses at each herbarium (those are the number of collections). Lines 148-149
- I have reinstalled the following sentences ‘Up to 60% of these herbarium specimens were imaged and measured for root lengths, rhizome internodes, petiole length, leaf length and width as well as the number of cross veins. When available, reproductive structure was measured and noted. Botanical drawings of reprehensive specimens were also made’ as requested. Lines 152-155
- Re “Transverse sections…” already in the previous revised ms. Now showing in Lines 170-172.
- Fig. 5. E is changed to be SE and Fig. 6. E is changed to be PE. Figure legends had been inserted “to show that H. stipulacea leaf has squamous appearance epidermal cells (SE); while H. balfourii leaf has pyramid appearance epidermal cells (PE)”. Lines 211-213
Spelling/typing errors:
1. Corrected reproductive Line 30
2. Corrected H. balfourii Line 54
3. Corrected area Line 70
4. Corrected papillose Lines 209,213
5. Corrected Die natürlichen Pflanzenfamilien Line 244
The other mentioned spelling/typing errors such as ‘Scanning, Materials, Mauritius, etc. had already been changed in the previous revised ms. It seems to me that there is a mixed up of the previous revised ms.
ps. I have tried to attach the latest revised ms here but failed to do so; instead a "author-coverletter-9246931.v1.pdf" appeared (see below). please ignor it. I believe you will be able to download it through the MDPI.
